# Origins of the central Macaronesian psyllid lineages (Hemiptera; Psylloidea) with characterization of a new island radiation on endemic *Convolvulus floridus* (Convolvulaceae) in the Canary Islands

**Saskia Bastin[1], J. Alfredo Reyes-Betancort[2], Felipe Siverio de la Rosa[1], Diana M. Percy[3]\***

**1** Instituto Canario de Investigaciones Agrarias, Unidad de Protección Vegetal, La Laguna, Tenerife, Spain, **2** Instituto Canario de Investigaciones Agrarias, Jardín de Aclimatación de La Oratava, Puerto de la Cruz, Tenerife, Spain, **3** Botany Department and Biodiversity Research Centre, University of British Columbia, Vancouver, British Columbia, Canada

* diana.percy@ubc.ca

## Abstract

A molecular survey of native and adventive psyllids in the central Macaronesian islands provides the first comprehensive phylogenetic assessment of the origins of the psyllid fauna of the Canary and Madeira archipelagos. We employ a maximum likelihood backbone constraint analysis to place the central Macaronesian taxa within the Psylloidea mitogenome phylogeny. The native psyllid fauna in these central Macaronesian islands results from an estimated 26 independent colonization events. Island host plants are predicted by host plants of continental relatives in nearly all cases and six plant genera have been colonized multiple times (*Chamaecytisus*, *Convolvulus*, *Olea*, *Pistacia*, *Rhamnus*, and *Spartocytisus*) from the continent. Post-colonization diversification varies from no further cladogenesis (18 events, represented by a single native taxon) to modest in situ diversification resulting in two to four native taxa and, surprisingly, given the diverse range of islands and habitats, only one substantial species radiation with more than four native species. Specificity to ancestral host plant genera or family is typically maintained during in situ diversification both within and among islands. Characterization of a recently discovered island radiation consisting of four species on *Convolvulus floridus* in the Canary Islands shows patterns and rates of diversification that reflect island topographic complexity and geological dynamism. Although modest in species diversity, this radiation is atypical in diversification on a single host plant species, but typical in the primary role of allopatry in the diversification process.

## Introduction

Macaronesia is considered an exemplary natural model for studying colonization and speciation in plant and arthropod lineages [1–4]. Species richness in this region, as with other

**Data Availability Statement:** "Data availability statement: All sequences generated for this study are deposited in Genbank (Accessions: OR068436-OR068551, OR859914-OR859951, OR864738-OR864739 (cox1), and OR067157-OR067192, OR863323-OR863358 (cytb)). Sequence alignment files and tree files are openly available from the figshare repository: https://doi.org/10.6084/m9.figshare.c.6985260.v1. All other data is contained in this article."

**Funding:** Saskia Bastin is recipient of a 2019-2023 PhD grant (TESIS2019010051) from the Agencia Canaria de Investigación Innovación y Sociedad de la Información (ACIISI), Consejería de Economía, Industria, Comercio y Conocimiento of the Gobierno de Canarias and the European Social Fund. This research was carried out with financial support from the research project CUARENTAGRI (MAC2/1.1a/231) awarded to Felipe Siverio de la Rosa. The funders had no role in study design, data collection and analysis, decision to publish, or preparation of the manuscript.

**Competing interests:** The authors have declared that no competing interests exist.

oceanic archipelagos, results from factors such as rates of colonization, extinction and in situ adaptive radiation that, in turn, are determined by dispersal ability, niche availability, climatic fluctuations, geological disturbance and geographical barriers [5–7]. Other influential aspects specific to each region such as local geology and climate history, and the extent of geographical isolation of individual islands, are also fundamental to understanding the evolution of species richness in each archipelago.

The Macaronesian biogeographical region encompasses four major archipelagos plus the smaller Selvagens archipelago, with the centrally positioned Canary Islands (seven islands), Madeira (two islands) and Selvagens (three islands) referred to as central Macaronesia. All archipelagos are considered of oceanic volcanic origin [8], but the number and placement of volcanic plumes responsible for the aerial islands remains debated, particularly for the complex geological parts of the Canarian archipelago [9,10]. In terms of biodiversity, the central Macaronesian archipelagos share several characteristics with other Pacific oceanic archipelagos, such as the Galapagos and Hawaiian Islands, including high endemism; e.g., in the Canary Islands, 45% of the arthropod fauna and 40% of the native vascular flora are endemic [11–13].

However, Macaronesian archipelagos differ in having a considerably wider geological age range, varying from 0.25 million years (Mya) for Pico (Azores) to 29 Mya for Selvagem Pequena Island (Selvagens Islands). This region also has a closer proximity to continental sources of colonization and, consequently, higher levels of immigration [1]. One of the easternmost of the Canary Islands, Fuerteventura, is only around 100 km from the northwest coast of the African continent and was even closer, around 65 km, during periods of glaciation [14–16]. Furthermore, the presence of seamounts located between islands and between archipelagos and the continent may have facilitated historic dispersal, both from the continent and between islands, serving as steppingstones during glacial periods when sea levels were lower [14,17,18]. It is therefore not surprising that most of the endemic Macaronesian flora and fauna has its closest relatives in nearby continental regions and a large number of Macaronesian endemics are shared between two or more archipelagos; the Canary Islands and Madeira being the islands sharing the largest number of endemic species [16,19,20]. Nevertheless, some groups display unusually disjunct distributions, for example, some elements of the flora and fauna have sister taxa in the east Mediterranean, Eurosiberia, East Asia, East Africa, South Africa, and the New World [16,21–24].

Psyllids or jumping plant-lice (Psylloidea) are a model system for studying island biogeography and evolutionary processes in conjunction with host plant selection as they are highly host specific phytophagous insects [25–28]. Most of the species feed on either one (monophagous) or a few related plant species (oligophagous) [28]. In general, studies have shown positive correlations between the diversity of phytophagous insects and host plants [29–32], with examples of insect speciation occurring in the absence of host plant diversification being rarer [33]. A common driver of speciation among phytophagous insects, including psyllids, is adaptation to different host plant species [28,34–37]. Most diversification in Macaronesian psyllids follows this typical process with speciation involving switching to closely related plant species [38], and hence, extent of in situ diversification in psyllid lineages partly depends on diversity in the host plant lineage. Similarly, successful colonization and establishment will depend on locating both familiar and unoccupied host plants [25,39]. Examples of phytophagous insect species radiations occurring on a single host plant species, particularly in sympatry, are rare [40–42]. A number of psyllid genera in continental regions are known to have multiple species occurring on the same host plant species (e.g., *Arytainilla*, *Bactericera*, *Cacopsylla*, *Calophya*, *Mitrapsylla*, *Queiroziella*). However, most of these examples have not explicitly tested whether the psyllid taxa sharing the same host are sister species and whether the speciation events may have occurred sympatrically or allopatrically. The recent description of an endemic psyllid

genus in the Canary Islands that has radiated on a single, endemic Canary Island plant, *Convolvulus floridus* (Convolvulaceae), provides an example of this less common scenario [43]. However, although atypical in occurring on a single host, the radiation of *Percyella* Bastin, Burckhardt & Ouvrard on *C. floridus* appears to be a textbook example of allopatric speciation, whereby each of the four psyllid species is found on *C. floridus* but on a separate island. This scenario also conforms to most records of closely related species that diversify on the same plant occurring in allopatry [38,44].

There are around 4000 described psyllid species worldwide [45] and they are found in all biogeographic realms except Antarctica, with their greatest diversity in tropical and south temperate regions [27,46]. Previous studies have revealed a wide taxonomic diversity of psyllid lineages in Madeira and the Canary Islands [47–51], and a recent survey described two new genera and 16 new species from the Canary Islands [43]. Of the 73 species recorded for the central Macaronesian islands [43,50], there are 58 native species in 17 genera in four of the seven recognized families [45] (Table 1). There is a high level of endemism that is particularly notable for the Canary Islands when compared with other Sternorrhyncha higher groups: <1%, 19%, 24%, 66% of the Aphidomorpha, Coccomorpha, Aleyrodidae, and Psylloidea are endemic respectively [13,43].

The total number of native Canarian psyllids is currently 51 species of which more than 80% are endemic (41 species) [43], while Madeira has 12 native psyllid species of which 50% are endemic [50] (Table 1). Four native species are currently considered to be central Macaronesian endemics and are found on both archipelagos. Two genera are endemic, the monotypic *Megadicrania* Loginova (Liviidae) found on Gran Canaria and Tenerife, and *Percyella* (Triozidae). The largest endemic radiation is represented by the legume-feeding *Arytinnis* Percy [38,47,51], with more modest in situ diversification found in three genera, *Arytaina* Foerster, *Drepanoza* Bastin, Burckhardt & Ouvrard and *Percyella* [43,47,51]. The majority of the native central Macaronesian taxa are found on the larger and more diverse archipelago of the Canary Islands (Table 1). The majority of the native Macaronesian species, including more than half the Canary Island native species (29 species in six genera) are in the family Psyllidae. Of the 62 species recorded from the Canary Islands, 11 are non-native and these are from nine genera (Table 2).

Few psyllids have been recorded from the low, drier, eastern islands, and most are considered non-native (three species) or native non-endemic (two species). In the case of historic records for two native legume-feeding species (*Arytaina devia* Loginova and *Arytinnis proboscidea* (Loginova) recorded from Fuerteventura [47], these are single individual records with no known host plants recorded on the island and therefore the records remain to be confirmed (indicated by [?] in Table 1). Only two native psyllid species have been recorded on Fuerteventura and Lanzarote, *Colposcenia viridis* Loginova and *Diaphorina continua* Loginova, despite host plants of other native psyllids occurring natively on all or some of the eastern islands (e.g., *Rhamnus crenulata*, *Convolvulus floridus*, *Gymnosporia cassinoides*, *Picconia excelsa*, *Pistacia atlantica* and *Olea cerasiformis*) [13]. Confirmed distributions for all endemic and most native species are therefore found only on the five higher elevation central and western islands, particularly the two central islands (Gran Canaria and Tenerife) where 15 of the 16 native genera representing 76% of the native species are recorded. The western islands (La Gomera, La Palma and El Hierro) have 43% of the native species in nine genera. Tenerife is both the largest island and the highest with an elevation of 3715 m. This island has the highest richness of native psyllids, with all genera except *Arytainilla* Loginova and *Spanioza* Enderlein represented, and around 60% of native psyllid species. Thereafter, in decreasing order, Gran Canaria, La Palma, La Gomera and El Hierro have 39%, 30%, 27% and 8% native species, respectively. All native species on the smallest and most westerly island of El Hierro also occur on La Palma. Five genera (*Agonoscena* Enderlein, *Drepanoza*, *Lisronia* Loginova, *Megadicrania*

**Table 1. Summary of the distribution, host plants and molecular data for native psyllid taxa (endemic species indicated by \*) of the central Macaronesian islands.**

| Species | Distribution | CA island | Recorded host plants in Macaronesia | Host plant family | Molecular data |
|---|---|---|---|---|---|
| **Family: Aphalaridae** | | | | | |
| *Agonoscena atlantica* Bastin, Burckhardt & Ouvrard, 2023 \* | CA | T | *Pistacia atlantica* | Anacardiaceae | cox1, cytb |
| *Agonoscena cisti* (Puton, 1882) | Western Palaearctic | C | No information | No information | cox1, cytb |
| *Agonoscena sinuata* Bastin, Burckhardt & Ouvrard, 2023 \* | CA | T | *Ruta pinnata* | Rutaceae | cox1, cytb |
| *Agonoscena targionii* (Lichtenstein, 1874) | Western Palaearctic | C | *Pistacia lentiscus* | Anacardiaceae | cox1, cytb |
| *Colposcenia viridis* Loginova, 1972 | Western Mediterranean, CA | C,F,L | Unknown | Unknown | – |
| *Lisronia echidna* Loginova, 1976 \* | CA | T,C | *Cistus monspeliensis* | Cistaceae | cox1, cytb |
| *Rhodochlanis salsolae* (Lethierry, 1874) | Western Mediterranean, MA | | *Suaeda vera* | Chenopodiaceae | – |
| **Family: Liviidae** | | | | | |
| *Euphyllura canariensis* Loginova, 1973 \* | CA, MA | P,T,C | *Picconia excelsa* | Oleaceae | cox1, cytb, mtg |
| *Euphyllura confusa* Bastin, Burckhardt & Ouvrard, 2023 \* | CA | T | *Olea europaea* | Oleaceae | cox1, cytb |
| *Euphyllura olivina* (Costa, 1839) | Mediterranean, CA, MA | C | *Olea europaea* | Oleaceae | cox1 |
| *Megadicrania tecticeps* Loginova, 1976 \* | CA | T,C | *Olea cerasiformis, Olea europaea* | Oleaceae | cox1, cytb |
| *Strophingia arborea* Loginova, 1976 \* | MA | | *Erica platycodon* spp. *maderincola* | Ericaceae | cox1, cytb, mtg |
| *Strophingia canariensis* Bastin, Burckhardt & Ouvrard, 2023 \* | CA | T | *Erica platycodon* spp. *platycodon* | Ericaceae | cox1, cytb |
| *Strophingia fallax* Loginova, 1976 \* | MA | | *Erica arborea* | Ericaceae | cox1, cytb |
| *Strophingia paligera* Bastin, Burckhardt & Ouvrard, 2023 \* | CA | P,G,T | *Erica canariensis* | Ericaceae | cox1, cytb |
| **Family: Psyllidae** | | | | | |
| *Arytaina devia* Loginova, 1976 \* | CA | G,T,F[?] | *Chamaecytisus proliferus* ssp. *angustifolius, Chamaecytisus proliferus* ssp. *proliferus* var. *palmensis* | Fabaceae | cox1, cytb |
| *Arytaina meridionalis* Bastin, Burckhardt & Ouvrard, 2023 \* | CA | C | *Chamaecytisus proliferus* ssp. *meridionalis* | Fabaceae | cox1, cytb |
| *Arytaina insularis* Loginova, 1976 \* | CA | P | *Chamaecytisus proliferus* ssp. *proliferus* var. *palmensis* | Fabaceae | cox1, cytb |
| *Arytaina nubivaga* Loginova, 1976 \* | CA | T | *Spartocytisus supranubius* | Fabaceae | cox1, cytb |
| *Arytaina vittata* Percy, 2003 \* | CA | H,P,G | *Spartocytisus filipes, Spartocytisus supranubius* | Fabaceae | cox1, cytb |
| *Arytainilla serpentina* Percy, 2003 \* | CA | P | *Spartocytisus filipes* | Fabaceae | mtg |
| *Arytinnis canariensis* Percy, 2003 \* | CA | T | *Teline canariensis* | Fabaceae | – |
| *Arytinnis diluta* (Loginova, 1976) \* | CA | T,C | *Teline canariensis, Teline microphylla* | Fabaceae | – |
| *Arytinnis dividens* (Loginova, 1976) \* | CA | P[?],G,T,C | *Chamaecytisus proliferus* | Fabaceae | mtg, cytb |
| *Arytinnis equitans* (Loginova, 1976) \* | CA | T,C | *Teline canariensis, Teline microphylla* | Fabaceae | mtg |
| *Arytinnis fortunata* Percy, 2003 \* | CA | P | *Teline splendens* | Fabaceae | – |
| *Arytinnis gomerae* Percy, 2003 \* | CA | G | *Teline stenopetala* ssp. *microphylla, Teline stenopetala* ssp. *pauciovulata* | Fabaceae | – |
| *Arytinnis hupalupa* Percy, 2003 \* | CA | G | *Teline stenopetala* ssp. *microphylla, Teline stenopetala* ssp. *pauciovulata* | Fabaceae | – |
| *Arytinnis incuba* (Loginova, 1976) \* | MA | | *Teline maderensis* | Fabaceae | – |

(*Continued*)

**Table 1.** (Continued)

| Species | Distribution | CA island | Recorded host plants in Macaronesia | Host plant family | Molecular data |
|---|---|---|---|---|---|
| *Arytinnis menceyata* Percy, 2003 * | CA | T | *Teline canariensis, Teline stenopetala* ssp. *spachiana* | Fabaceae | – |
| *Arytinnis modica* (Loginova, 1976) * | CA | H,P | *Chamaecytisus proliferus, Teline stenopetala* ssp. *microphylla* | Fabaceae | – |
| *Arytinnis nigralineata* (Loginova, 1976) * | CA | G,T,C | *Adenocarpus foliolosus* | Fabaceae | cox1 |
| *Arytinnis occidentalis* Percy, 2003 * | CA | H,P | *Teline stenopetala* ssp. *microphylla, Teline stenopetala* ssp. *sericea, Teline stenopetala* ssp. *stenopetala* | Fabaceae | – |
| *Arytinnis ochrita* Percy, 2003 * | CA | T | *Teline osyroides* ssp. *osyroides* | Fabaceae | – |
| *Arytinnis pileolata* (Loginova, 1976) * | CA | T | *Teline canariensis, T. osyroides* ssp. *sericea, T. stenopetala* ssp. *spachiana* | Fabaceae | – |
| *Arytinnis proboscidea* (Loginova, 1976) * | CA | P,G,T, C, F[?] | *Adenocarpus foliolosus, Adenocarpus viscosus* | Fabaceae | mtg |
| *Arytinnis prognata* (Loginova, 1976) * | CA | C | *Teline microphylla* | Fabaceae | – |
| *Arytinnis romeria* Percy, 2003 * | CA | C | *Teline rosmarinifolia* | Fabaceae | – |
| *Arytinnis umbonata* (Loginova, 1976) * | MA | | *Genista tenera* | Fabaceae | – |
| *Cacopsylla atlantica* (Loginova, 1976) * | CA, MA | P,G,T,C | *Salix canariensis* | Salicaceae | cox1, cytb |
| *Cacopsylla crenulatae* Bastin, Burckhardt & Ouvrard, 2023 * | CA | G,T | *Rhamnus crenulata* | Rhamnaceae | cox1, cytb |
| *Cacopsylla exima* (Loginova, 1976) * | CA, MA | T | *Rhamnus glandulosa* | Rhamnaceae | cox1 |
| *Cacopsylla falcicauda* Bastin, Burckhardt & Ouvrard, 2023 * | CA | G | *Rhamnus glandulosa* | Rhamnaceae | cox1, cytb |
| *Diaphorina continua* Loginova, 1972 | Western Mediterranean, CA | F,L | Unknown | Unknown | – |
| *Diaphorina gonzalezi* Bastin, Burckhardt & Ouvrard, 2023 * | CA | T | *Gymnosporia cassinoides* | Celastraceae | cox1, cytb, mtg |
| *Livilla monospermae* Hodkinson, 1990 * | CA | H,P,G,T | *Retama rhodorhizoides* | Fabaceae | mtg |
| **Family: Triozidae** | | | | | |
| *Drepanoza canariensis* Bastin, Burckhardt & Ouvrard, 2023 * | CA | T | *Convolvulus canariensis* | Convolvulaceae | cox1 |
| *Drepanoza fernandesi* (Aguiar, 2001) * | MA | | *Pittosporum coriaceum* | Pittosporaceae | cox1, cytb |
| *Drepanoza fruticulosi* Bastin, Burckhardt & Ouvrard, 2023 * | CA | T | *Convolvulus fruticulosus* | Convolvulaceae | cox1, cytb |
| *Drepanoza molinai* Bastin, Burckhardt & Ouvrard, 2023 * | CA | T | *Withania aristata* | Solanaceae | cox1, cytb |
| *Drepanoza montanetana* (Aguiar, 2001) * | CA | C | Unknown | Unknown | – |
| *Drepanoza pittospori* (Aguiar, 2001) * | MA | | *Pittosporum coriaceum* | Pittosporaceae | cox1, cytb |
| *Lauritrioza laurisilvae* (Hodkinson, 1990) * | CA, MA | P,G,T,C | *Laurus* spp. | Lauraceae | cox1, cytb |
| *Percyella benahorita* Bastin, Burckhardt & Ouvrard, 2023 * | CA | P | *Convolvulus floridus* | Convolvulaceae | cox1, mtg |
| *Percyella canari* Bastin, Burckhardt & Ouvrard, 2023 * | CA | T,C | *Convolvulus floridus* | Convolvulaceae | cox1 |

(*Continued*)

**Table 1.** (Continued)

| Species | Distribution | CA island | Recorded host plants in Macaronesia | Host plant family | Molecular data |
|---|---|---|---|---|---|
| *Percyella gomerita* Bastin, Burckhardt & Ouvrard, 2023 * | CA | G | *Convolvulus floridus* | Convolvulaceae | cox1, cytb |
| *Percyella guanche* Bastin, Burckhardt & Ouvrard, 2023 * | CA | P,T | *Convolvulus floridus* | Convolvulaceae | cox1, cytb |
| *Spanioza* sp. [cf. *coquempoti* Burckhardt & Lauterer, 2006] | CA | C | No information | No information | – |

Abbreviations: CA: Canary Islands, MA: Madeira; H: El Hierro, P: La Palma, G: La Gomera, T: Tenerife, C: Gran Canaria, F: Fuerteventura, L: Lanzarote; [?] indicates island distribution needs confirmation. Molecular data generated during this study: cox1: cytochrome oxidase 1, cytb: cytochrome B, mtg: data from Percy et al. [52]. Host plants given are records from Macaronesia if known.

**Table 2. Summary of the distribution, host plants and molecular data for the non-native psyllid taxa of the central Macaronesian islands.**

| Species | Archipelago | CA island | Recorded host plants in Macaronesia | Host plant family | Molecular data |
|---|---|---|---|---|---|
| **Family: Aphalaridae** | | | | | |
| *Ctenarytaina eucalypti* (Maskell, 1890) | CA, MA | T | *Eucalyptus globulus*, *Eucalyptus* sp. | Myrtaceae | cox1, cytb |
| *Ctenarytaina spatulata* Taylor, 1997 | MA | | *Eucalyptus globulus* | Myrtaceae | mtg |
| *Glycaspis brimblecombei* Moore, 1964 | CA | T,C | *Eucalyptus* sp. | Myrtaceae | cox1, cytb |
| **Family Carsidaridae** | | | | | |
| *Macrohomotoma gladiata* Kuwayama, 1908 | CA | T,L | *Ficus microcarpa* | Moraceae | cox1, cytb, mtg |
| **Family: Psyllidae** | | | | | |
| *Acizzia acaciaebaileyanae* (Froggatt, 1901) | CA | P | No information | No information | – |
| *Acizzia uncatoides* (Ferris & Klyver, 1932) | CA, MA | H,P,G,T,F,L | *Acacia baileyana, A. cyclops, A. longifolia, A. mearnsii, Paraserianthes lophantha* | Fabaceae | cox1, cytb, mtg |
| *Cacopsylla fulguralis* (Kuwayama, 1908) | MA | | *Elaeagnus pungens* | Elaeagnaceae | – |
| *Cacopsylla pyri* (Linnaeus, 1758) | MA | | *Pyrus communis* | Rosaceae | mtg |
| *Heteropsylla cubana* Crawford, 1914 | CA | G,T | *Leucaena leucocephala* | Fabaceae | cox1, cytb, mtg |
| *Platycorypha nigrivirga* Burckhardt, 1987 | CA, MA | T | *Tipuana tipu* | Fabaceae | cox1, cytb |
| **Family Triozidae** | | | | | |
| *Bactericera tremblayi* (Wagner, 1961) | CA | T | *Allium cepa, Allium ampeloprasum* var. *porrum* | Liliaceae | cox1, cytb |
| *Bactericera trigonica* Hodkinson, 1981 | CA | T,C | *Daucus carota* | Apiaceae | cox1, mtg |
| *Heterotrioza chenopodii* (Reuter, 1876) | CA, MA | H,P,G,T,C, F,L | *Chenopodium album, C. murale, Chenopodium* spp. | Amaranthaceae | cox1, cytb, mtg |
| *Trioza erytreae* (Del Guercio, 1918) | CA, MA | H,P,G,T,C | *Citrus* spp. | Rutaceae | cox1, cytb, mtg |
| *Trioza urticae* (Linnaeus, 1758) | MA | | *Urtica* sp. | Urticeae | mtg |

Abbreviations: CA: Canary Islands, MA: Madeira; H: El Hierro, P: La Palma, G: La Gomera, T: Tenerife, C: Gran Canaria, F: Fuerteventura, L: Lanzarote. Molecular data: cox1: cytochrome oxidase 1, cytb: cytochrome B, mtg: obtained from Percy et al. [52]. Host plant records in Macaronesia (all records from Canary Islands) are given if known.

Loginova, and S*panioza*) are only found on the central islands. Eight genera (*Arytaina*, *Arytinnis*, *Cacopsylla* Ossiannilsson, *Euphyllura* Foerster, *Lauritrioza* Conci & Tamanini, *Livilla* Curtis, *Percyella*, and *Strophingia* Enderlein) are found on both central and western islands, but only *Arytaina* and *Arytinnis* are present on all five islands. Lastly, *Arytainilla* has only been recorded from the western island of La Palma. Only four native species are recorded from both the Canary and Madeira archipelagos (*Cacopsylla atlantica* (Loginova), *C. exima* (Loginova), *Euphyllura canariensis* Loginova, and *Lauritrioza laurisilvae* (Hodkinson)), and these species all appear to be occupying the same host niche on both archipelagos [43].

The objectives of this study are: 1) establish the phylogenetic placement of central Macaronesian lineages within the superfamily Psylloidea; 2) estimate the number of colonization events required to explain the diversity of psyllids in these islands; 3) interpret patterns of island–continent host plant associations; and 4) characterize the biogeographic and population structure in the radiation of *Percyella* on *Convolvulus floridus*.

## Materials and methods

### Sampling

Our molecular survey represents all native psyllid lineages in the central Macaronesian islands except *Rhodochlanis* Loginova (Table 1). In addition to the seven lineages (nine species) represented in Percy et al. [52], we sampled 35 of the 58 native species found in central Macaronesia. Of the 51 native species in the Canary Islands, the only species not in our sampling are *Colposcenia viridis*, *Diaphorina continua*, *Drepanoza montanetana* (Aguiar), the undescribed species of *Spanioza*, and 15 of the 18 species from the *Arytinnis* radiation (analyzed previously in Percy 2003b [38]); *Agonoscena cisti* (Puton) and *Euphyllura olivina* (Costa) although not sampled from the Canary Islands were sampled from Spain. Twenty-one species (12 native and nine adventive) are recorded from Madeira [50], and all but three of the native species are represented in our data. In addition, we sampled or included data for 13 of the 15 confirmed adventive Macaronesian species (Table 2). Specimens were either field collected by SB during this study or obtained from previous collections made by DP and others; field sampling was performed from 1997–2023. Field sampled material was collected by sweep netting or aspirated directly from the host plant; specimens were then transferred live into 90–95% ethanol and stored at -20°C. In a few cases, if adults were not observed on the host plant, leaves with either immatures or galls were transferred to a plastic box and kept at room temperature until adult emergence. Host plant associations in most cases were confirmed by observation of last stage immatures [43]. Tables 1 and 2 show details of all taxa recorded from central Macaronesia and whether molecular data was obtained. Details of non-Macaronesian taxa used in the molecular analyses are given in S1 Table. Three continental taxa, *Drepanoza lienhardi* (Burckhardt), *Cacopsylla alaterni* (Foerster) and *C. myrthi* (Puton), were sampled from dry pinned material provided by Daniel Burckhardt (Naturhistorisches Museum, Basel); and a specimen of *Strophingia ericae* (Curtis) (JHM6411) collected in 1994 was obtained from the Natural History Museum, UK; the sequence for *Agonoscena pistaciae* Burckhardt & Lauterer was obtained from GenBank KP192847, and sequence data for an undescribed *Agonoscena* sp. from Greece was obtained from Arthemis Database (INRAE-CBGP) (Specimen code: CCOC11846_0202).

Two of the *Percyella* taxa (*P. guanche* Bastin, Burckhardt & Ouvrard and *P. benahorita* Bastin, Burckhardt & Ouvrard) from the radiation on *Convolvulus floridus* were initially collected in the Canary Islands by DP in 1998, two additional taxa were discovered during extensive sampling across Gran Canaria, Tenerife, La Gomera and La Palma by SB between 2018 and 2022. *Convolvulus floridus* was also surveyed in Lanzarote, but no adults, immatures or galls were found. *Percyella* specimens were collected from five sites in La Gomera, eight in La

Palma, eight in Tenerife and three in Gran Canaria. Site locations are detailed in Table 3. Additionally, all nine endemic species of *Convolvulus* in the Canary Islands were surveyed (see S2 Table) to confirm distribution of psyllids on this host genus.

Host plants in the Canary and Madeira islands, if known, are given in Tables 1 and 2. The host plant taxonomy follows that given in WFO [53] for Madeira species and Gobierno de Canarias [13] for Canary Island species. The taxonomy of genistoid legumes is in flux (e.g., with regard to recognition of a paraphyletic *Teline* nested within *Genista* sensu lato [54], and different species and subspecies epithets used for *Chamaecytisus proliferus* sensu lato) and in these cases we have elected to follow taxonomic names preferentially used in Macaronesia.

## Molecular procedures and sequencing

Non-destructive DNA extraction protocols using whole individual specimens were performed with either the Chelex protocol (following Casquet et al. [55]) or the Qiagen Blood and Tissue Kit (Qiagen) (following Percy et al. [52]). DNA voucher specimens were preserved in ethanol (70–85%) and deposited in Instituto Canario de Investigaciones Agrarias (Valle de Guerra, Spain) or retained in DPs personal collection (DMPC, University of British Columbia). The mitochondrial fragment of cytochrome oxidase subunit 1 (cox1) was amplified for the majority of taxa using either primer pairs, LCOP-F and HCO2198 (length 658 bp), or mtd6 and H7005P-R (length 850 bp) (see Bastin et al. [56]). In a few cases, specimens that failed to amplify with these two primer pairs (<5% of specimens) were amplified with tRWF1 and LepR1 (length ~860 bp) or mtd6 and mtd9 (length 472 bp) [52,56]. All primer details, combinations and references are given in Table 4. An additional mitochondrial fragment of cytochrome B (cytb, length 385 bp) was amplified with primers cytBF and cytBR [52]. For most of the species, these gene regions were obtained from the same individual, but in a few cases from different individuals from the same collection event. In three instances, sequences were obtained from different populations on the same island and in only one instance (for *Megadicrania tecticeps* Loginova) were sequences obtained from populations on different islands. PCR amplification was performed in a 25 μl final reaction volume containing 0.4 μM of each primer, 3 mM $MgCl_2$, $NH_4$ buffer (1×), 0.2 mM of each dNTP, 0.4 mg/ml of acetylated bovine serum albumin (BSA), 0.02 unit/μl of Taq-polymerase (Bioline) and 2 μl of DNA extract (concentration not determined). Polymerase chain reactions (PCRs) were carried out in Swift™ Maxi Thermal Cyclers (ESCO Technologies) applying the following thermal step: initial denaturation for 4 min at 94°C, followed by 39 cycles of 30 s at 94°C, 30 s at annealing temperature of 50–56°C (see Table 4), and 45 s at 72°C, with a final extension step of 10 min at 72°C. PCR products were enzymatically cleaned with 0.025 unit/μl rApid alkaline phosphatase (Roche) and 50 unit/ml exonuclease I (BioLabs) for 15 min at 37°C followed by 15 min at 95°C. The purified products were sequenced in both directions at Macrogen Inc. (Madrid, Spain) or Eurofins (Kentucky, USA). Additional PCR amplifications, including using older dry material (> 10 years old), were performed following protocols described in Percy et al. [52]. Sequences were checked, edited, and assembled with CLUSTALW [57] within the MEGA 7 software [58]. DNA sequences generated in this study are deposited in GenBank with accession numbers OR068436-OR068551, OR859914-OR859951, OR864738-OR864739 (cox1), and OR067157-OR067192, OR863323-OR863358 (cytb) and additional Genbank accessions are given in Percy [41], Percy et al. [52] and Bastin et al. [56].

## Phylogenetic analyses

Phylogenetic analysis to place 60 species (both Macaronesian and outgroup) within the Psylloidea tree employed a maximum likelihood (ML) constraint analysis run with RAxML (v.

**Table 3. Summary of the specimens of four *Percyella* species used in the phylogenetic and haplotype analyses.**

| Species (% max div.) | Specimen ID (# indiv.) | Coll. date | Is. | Site | GPS | Elev.(m) | Region |
|---|---|---|---|---|---|---|---|
| *P. guanche* (3.1%) | PN118 (2), PN119 | 20 March 2020 | T | T5 | 28.521110, -16.335329 | 410 | Tegueste |
| | PN140, PN141 | 01 March 2022 | T | T1 | 28.369439, -16.849553 | 130 | Buenavista del Norte |
| | PN142, PN143, PN144 | 01 March 2022 | T | T2 | 28.366387, -16.774888 | 320 | El Tanque |
| | PN146 | 01 March 2022 | T | T3 | 28.373472, -16.745087 | 70 | El Guincho |
| | PN147, PN148 | 06 March 2022 | T | T6 | 28.565124, -16.214306 | 100 | Taganana |
| | PN149, PN150 | 16 March 2022 | T | T7 | 28.515867, -16.176825 | 130 | Las Gaviotas |
| | PN151, PN152 | 16 March 2022 | T | T8 | 28.552697, -16.343148 | 64 | Bajamar |
| | PN97 | 08 January 2022 | P | P9 | 28.6493135, -17.9015229 | 390 | El Paso |
| | PN98 | 08 January 2022 | P | P9 | 28.6493135, -17.9015229 | 390 | El Paso |
| *P. benahorita* (4.3%) | PN85, PN86 | 07 January 2022 | P | P1 | 28.6884804, -17.7661645 | 50 | Santa Cruz de La Palma |
| | PN87, PN88 | 07 January 2022 | P | P2 | 28.8119329, -17.7797713 | 120 | Barlovento |
| | PN89 | 07 January 2022 | P | P3 | 28.8317471, -17.7994751 | 470 | Barlovento |
| | PN91 | 07 January 2022 | P | P4 | 28.705665, -17.7569 | 80 | Santa Cruz de La Palma |
| | PN92 | 08 January 2022 | P | P5 | 28.66548, -17.76926 | 20 | Breña Alta |
| | PN93 | 08 January 2022 | P | P6 | 28.660419, -17.792220 | 350 | Breña Alta |
| | PN94, PN95, PN96 | 08 January 2022 | P | P8 | 28.660200, -17.935223 | 230 | Los Llanos de Aridane |
| | DP195-98 | 18 May 1998 | P | DP195 | 28.658333, -17.933333 | 200 | Barranco de las Angustias |
| *P. gomerita* (0.9%) | PN99, PN100, PN101, PN102, PN103, PN104 | 15 January 2022 | G | G1 | 28.184799, -17.193395 | 260 | Agulo |
| | PN105 | 15 January 2022 | G | G2 | 28.148807, -17.193500 | 310 | Hermigua |
| | PN106, PN107, PN108, PN109 | 15 January 2022 | G | G3 | 28.1932796, -17.1977888 | 240 | Agulo |
| | PN110, PN111 | 16 January 2022 | G | G4 | 28.179217, -17.262758 | 210 | Vallehermoso |
| | PN112 | 16 January 2022 | G | G7 | 28.061213, -17.225814 | 670 | Alajeró |
| *P. canari* (0.6%) | PN113, PN134, PN135 | 19 February 2022 | C | C1 | 28.135944, -15.580556 | 130 | Moya |
| | PN136, PN137 | 19 February 2022 | C | C2 | 28.125750, -15.568167 | 260 | Moya |
| | PN138, PN139 | 19 February 2022 | C | C3 | 28.037583, -15.458306 | 560 | Telde |
| | PN1 | 18 June 2018 | T | T4 | 28.492267, -16.329015 | 570 | San Cristóbal de La Laguna |

(*Continued*)

**Table 3.** (Continued)

| Species (% max div.) | Specimen ID (# indiv.) | Coll. date | Is. | Site | GPS | Elev.(m) | Region |
|---|---|---|---|---|---|---|---|
| | PN4 | 03 December 2018 | T | T4 | 28.492267, -16.329015 | 570 | San Cristóbal de La Laguna |

Abbreviations: % max div.: maximum intraspecific cox1 divergence (uncorrected p-distances); Island: Is., P: La Palma, G: La Gomera, T: Tenerife, C: Gran Canaria. Sites are shown in Figs 2 and 3. Variation in sequence length obtained for all *Percyella* specimens is shown in (S1–S3 Figs).

8.2.12) [64] on the CIPRES Science Gateway [65]. The constraint tree employed was the total evidence tree obtained from mitogenome data presented in Percy et al. [52]. The constraint tree option allows the user to specify an incomplete multifurcating constraint tree for the RAxML search. Initially, multifurcations were resolved randomly and the additional taxa were added using a maximum parsimony criterion to compute a comprehensive (containing all taxa) bifurcating tree [64]. This tree was then further optimized under ML criteria respecting the given constraints with the added taxa unconstrained (i.e., can be placed in any part of the tree). Data partitions were specified for codon position and RNA regions, and ML search criteria employed model GTRCAT, 1000 rapid bootstraps, and Gamma optimization of tree space. In order to maximize the power of this method to place shorter sequences correctly within the Psylloidea phylogeny, we used both cox1 and cytb regions where available.

To further investigate placement of taxa and relationship to outgroups for genera *Drepanoza* and *Percyella*, we selected related outgroup taxa from within Group A in Percy et al. [52] (Table 3). For *Percyella* species, multiple sites were surveyed across four islands (Table 3). We used a cox1 dataset (1216 bp in length, 419 variable characters of which 344 are parsimony informative, 105 informative within *Drepanoza*, and 171 informative within *Percyella*) and performed three phylogenetic analyses using neighbor-joining (NJ) with PAUP* v4.0a [66], maximum likelihood (ML) with RAxML (with the same parameters as specified for the Psylloidea analysis but minus a tree constraint), and Bayesian inference (BI) using the BEAST v2.7.3 package [67]. Three identical sequences in *Percyella* were removed from the dataset used for ML and BI analyses. The NJ method with all 50 sequences was performed with uncorrected (p) distances in PAUP* [66], and clade support was obtained with a NJ bootstrap analysis (1000 replicates), this method was also used to estimate the maximum intraspecific genetic divergences reported in Table 3. The ML and BI analyses used specified data partitions for codon position and the noncoding tRNA-W region. Five additional taxa from the Hawaiian *Pariaconus* Enderlein radiation were added for the Bayesian dating analyses to test consistency across

**Table 4. Primer combinations used to amplify cox1 with reference, sequence, annealing temperature, and amplicon length.**

| Primer | Reference | Direction | Sequence (5′-3′) | Tm (˚C) | Amplicon lenght (bp) |
|---|---|---|---|---|---|
| mtd6 (C1-J-1718) | [59] | Forward | GGAGGATTTGGAAATTGATTAGTTCC | 50 | 472 |
| mtd9 (C1-N-2191) | [59] | Reverse | CCCGGTAAAATTAAAATATAAACTTC | | |
| mtd6 (C1-J-1718) | [59] | Forward | GGAGGATTTGGAAATTGATTAGTTCC | 50 | 850 |
| H7005P-R | [60] | Reverse | TGAGCTACTACRTARTATGTRTCATG | | |
| LCOP-F | [56] | Forward | AGAACWAAYCATAAAAYWATTGG | 48 | 658 |
| HCO2198 | [61] | Reverse | TAAACTTCAGGGTGACCAAAAAATCA | | |
| tRWF1 | [62] | Forward | AACTAATARCCTTCAAAG | 50 | ±860 |
| LepR1 | [63] | Reverse | TAAACTTCTGGATGTCCAAAAAATCA | | |

Abbreviation: Tm: primer melting temperature.

the calibrated tree with island ages. For the dating analyses, in addition to the four data partitions, substitution rates were estimated, Gamma count set to 4, and substitution model set to HKY with empirical frequencies; a strict clock model was used as the data are from a single mitochondrial region and assumed to be clock-like; tree prior used Calibrated Yule Model with Uniform birthRate parameter (using Gamma birthRate parameter produced near identical results). To calibrate the trees, we used internal calibration nodes with monophyly enforced as MRCA (most recent common ancestor) priors. Four internal node calibration priors were used, three of these in the Hawaiian *Pariaconus* radiation where sister taxon pairs on three islands had previously been studied (on Kauai, Oahu, and Hawaii islands) [41,52], the fourth calibration was La Palma island which has a comparably well defined geological age within the Canaries (whereas older islands in the Canaries have more complex geological histories and a wider age range of geological formations). In addition, La Palma was found to have the highest haplotype diversity within *Percyella*. Age calibration analyses were run in the following combinations: using only La Palma, using only Kauai, using all three Hawaiian Islands. We set the age priors using a normal distribution and set the mean so that the 95% upper range was at the maximum geological island age. Thus, island calibration priors were set as follows: La Palma with a mean age of 1.8 Mya (million years ago) and 95% range of 1.64–1.96 Mya; Kauai with a mean age of 4.8 Mya and 95% range of 4.64–4.96 Mya; Oahu with a mean age of 3.5 Mya and 95% range of 3.34–3.66 Mya; Hawaii with a mean age of 0.8 Mya and 95% range of 0.64–0.96 Mya. We used a MCMC (Markov chain Monte Carlo) chain length of 25 million, tracelog and treelog set to 1000, and a 10% burnin. Tracer v1.7.2 [68] was used to check chain convergence and ESS (effective sample size) values. Increasing the MCMC chain length from 10 million to 25 million was required to obtain satisfactory ESS values ($> 500$) for all parameters. The age calibrated tree was visualized with FigTree v1.4.4 [69] showing 95% HPD (highest posterior density interval) bars on the nodes.

To further explore and characterize biogeographic and population level patterns in the *Percyella* radiation of four closely related species, haplotype variation in the four species across the four islands was analysed using PopART v1.7 [70]. The 50 cox1 *Percyella* sequences were used to create a haplotype median-joining network [71] and haplotype map. Geotags for each sequence enabled geographic clustering by the k-means algorithm, with a centroid georeference for each island. Basic population structure was assessed from a simple AMOVA (analysis of molecular variation) as a proportion of nucleotide diversity between and within populations of the four species. Due to variation in sequence length across the 50 *Percyella* sequences as a result of amplification with different primer combinations, the haplotype analysis was performed with only the 280 bp shared across all individuals, whereas the NJ (and genetic distances reported), ML, and BI analyses used the full-length sequences.

## Results

### Phylogenetic placement of Macaronesian lineages within Psylloidea and estimated number of colonization events

The ML backbone constraint analysis provides a best estimate of the phylogenetic placement of the central Macaronesian taxa in the broader Psylloidea phylogeny. Placement of the native psyllid genera indicates that central Macaronesian lineages are distributed throughout the Psylloidea tree (Fig 1). With reference to the groups determined by Percy et al. [52], native Macaronesian taxa are in families: Triozidae, within Group A (*Drepanoza*, *Percyella*), and Group B (*Lauritrioza laurisilvae*); Psyllidae, within Group O (*Cacopsylla*), Group P (*Arytaina*, *Arytainilla*, *Arytinnis*, *Livilla*), and Group BB (*Diaphorina gonzalezi* Bastin, Burckhardt & Ouvrard); Liviidae, within Group FF (*Euphyllura*, *Megadicrania tecticeps*, *Strophingia*); and Aphalaridae

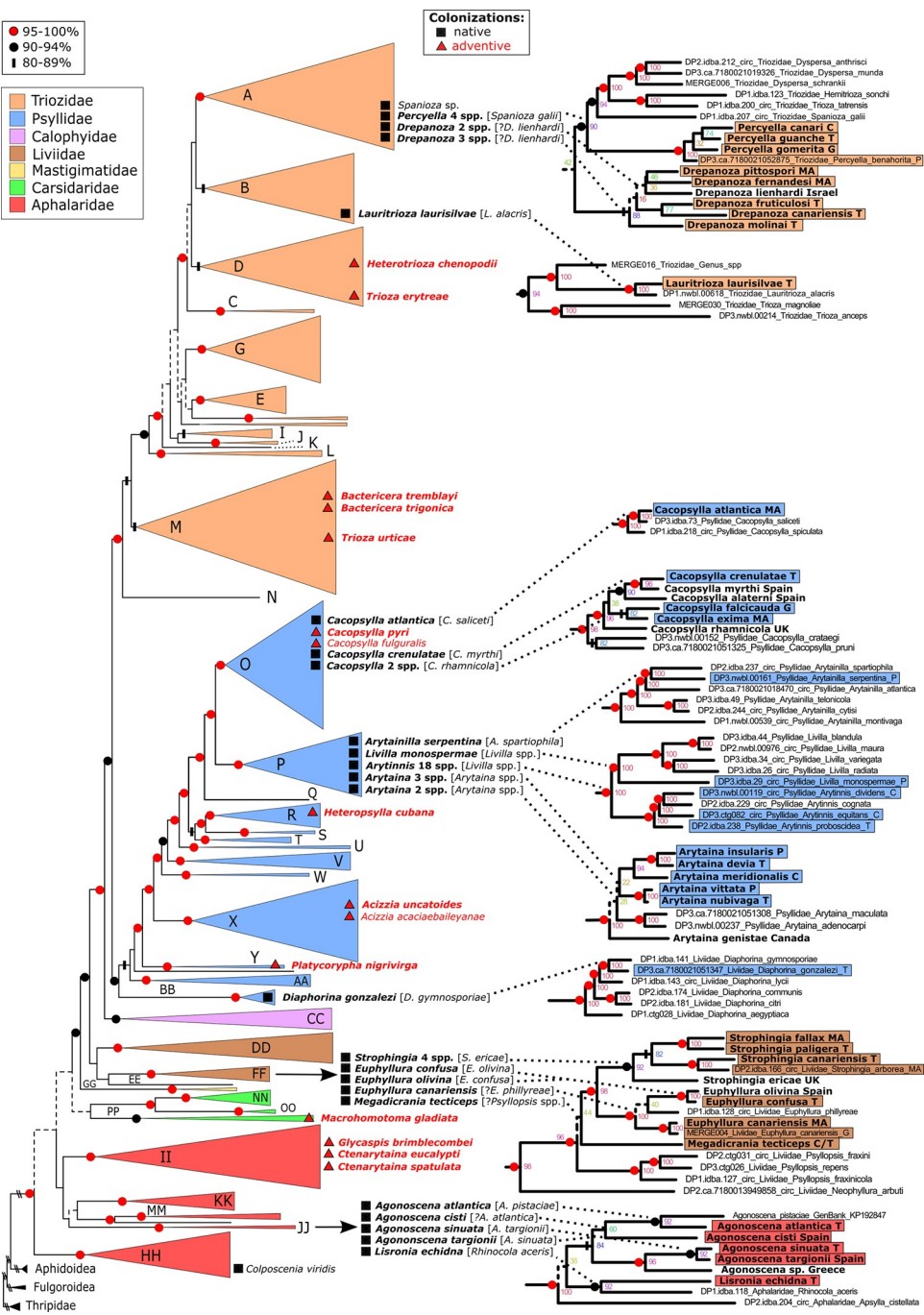

**Fig 1. Maximum likelihood analysis of the superfamily Psylloidea using the mitogenome data published in Percy et al. [52] as a backbone tree constraint and additional sequences from this study (see text, Tables 1–3).** The estimated number of independent colonizations/introductions for native and adventive taxa in the central Macaronesian islands is indicated by square/triangle symbols. For systematic placement, names in bold indicate sequence data was used in our analysis, and non-bold indicates no sequence data available so placement inferred by congeneric taxa; phylogenetically closest taxon in [] for well supported, or [?] if support is less than 80% bootstrap. Insets show taxon placement for native Macaronesian taxa (box colour indicates psyllid family); taxon names in bold indicate short sequences placed in the mitogenome backbone phylogeny.

within Group JJ (*Agonoscena, Lisronia echidna* Loginova). Within these major groups, placements within subgroups are moderately to well supported (with bootstrap >80%), with the exception of *Drepanoza* where placement of this genus is topologically stable but remains unconfirmed due to lack of bootstrap support (Fig 1). The placement of *Drepanoza* near *Percyella* and *Spanioza* within Group A is also supported morphologically [43].

The native psyllid fauna of the central Macaronesian islands has resulted from an estimated 26 independent colonization events (Fig 1). More than half of these colonizations, 18, are represented by a single native taxon, in other words no further cladogenesis. Three of the colonization events resulted in limited cladogenesis (with two native sister taxa); and five colonization events resulted in further cladogenesis that can be characterized as either modest or substantial species radiations. The colonization event that gave rise to the genus *Arytinnis* represents the only large psyllid radiation with 18 endemic Macaronesian species [38,51]. Four other groups have undergone modest radiations. One of these is exclusively in the Canary Islands: *Percyella* with four species, each native to a different island (further explored below). Two groups are interpreted as colonizing Macaronesia twice: *Arytaina* with two or three native species resulting per colonization (Fig 1), and *Drepanoza* with two or three native species per archipelago (Figs 1 and 2). The placement of the only continental species in *Drepanoza* (*D. lienhardi*) among the island species is not straightforward to interpret at this time due to lack of resolution; an alternative scenario to multiple colonizations of Macaronesia by *Drepanoza* is a back colonization from island to continent. There is a potential Macaronesian radiation of four species of *Strophingia*, but the pattern is unusual as sister taxon pairs are found on different archipelagos, requiring two cladogenic events between the Canaries and Madeira: *Strophingia fallax* Loginova from Madeira groups strongly (100%) with *Strophingia paligera* Bastin, Burckhardt & Ouvrard from the Canary Islands, and *Strophingia arborea* Loginova from Madeira groups strongly (100%) with *Strophingia canariensis* Bastin, Burckhardt & Ouvrard from the Canary Islands. However, morphological evidence suggests *Strophingia paligera* may be close to a continental species, *S. cinerea* Hodkinson, while the immature morphology of *S. canariensis* is most similar to that of another continental species, *S. proxima* Hodkinson [43]. All species feed on *Erica* spp. and the continental species occur in the Western Mediterranean but are not sampled in our analysis. It is therefore possible that multiple colonizations account for *Strophingia* in Macaronesia, but this remains to be tested. The only other interarchipelago cladogenic event recorded is in a previous study of *Arytinnis*, where the two Madeira species are nested within the Canary Island radiation, but in this case, it is uncertain whether the cladogenic event was via colonization directly from the Canary Islands to Madeira, or via a back colonization to the continent [38].

Of the four species that occur in both Canary and Madeira archipelagos, we only included samples from both archipelagos for *Euphyllura canariensis* and *Cacopsylla atlantica* (Fig 1). In both cases, intraspecific genetic divergence was moderately high between archipelagos (cox1: 3.5% and 2.0% respectively) indicating that both species have non-interbreeding and diverging populations on these archipelagos; some morphological variation between archipelagos was observed for *E. canariensis* but not as to support recognition of separate sister species [43].

The 15 adventive species recorded for central Macaronesia are from eight major phylogenetic groups (Fig 1). A number of these taxa are worldwide invasives encountered in many different parts of the world (e.g., *Acizzia uncatoides* (Ferris& Klyver), *Cacopsylla fulguralis* (Kuwayama), *Ctenarytaina eucalypti* (Maskell), *Glycaspis brimblecombei* Moore, *Heteropsylla cubana* Crawford, and *Macrohomotoma gladiata* Kuwayama). Others are more localized introductions from Europe and Mediterranean regions (e.g., the *Bactericera* Puton spp., *Cacopsylla pyri* (Linnaeus), *Heterotrioza chenopodii* (Reuter), and *Trioza urticae* (Linnaeus)) (see Bastin et al. [43]).

## Insights from host plant associations

Our backbone constraint analysis (Fig 1) provides evidence for, and in most cases confirmation of systematic placement of all included island taxa. This phylogenetic framework provides insights on host plant associations by comparing host associations in closely related species, as follows:

*Lauritrioza laurisilvae* on *Laurus novocanariensis* groups strongly with a continental Lauraceae-feeding species, *Lauritrioza alacris* (Flor), found on *Laurus nobilis*.

The three endemic *Cacopsylla* species on *Rhamnus* hosts represent two independent colonization events, and all three species group with continental species on *Rhamnus*; the endemic *Cacopsylla crenulatae* Bastin, Burckhardt & Ouvrard on *Rhamnus crenulata* groups with strong support (96%) with continental *C. myrthi*, and these group with continental *C. alaterni* (90%). The other endemic *Cacopsylla*, *C. falcicauda* Bastin, Burckhardt & Ouvrard and *C. exima*, both on *Rhamnus glandulosa*, are only moderately supported (82%) as sister taxa, and all of these *Rhamnus*-feeding *Cacopsylla* together group with a continental species on *Rhamnus*, *C. rhamnicola* (Scott) (96%). The fourth Macaronesian endemic *Cacopsylla*, *C. atlantica*, is found on *Salix canariensis*, a Macaronesian endemic tree, and is strongly supported (100%) as grouping with a continental species, *C. saliceti* (Foerster) on *Salix* spp. hosts.

Of the genistoid legume-feeding group, *Arytainilla serpentina* Percy on *Spartocytisus filipes* is strongly supported (100%) grouping with a continental species, *A. spartiophila* (Foerster), on host *Cytisus scoparius*. The five *Arytaina* species represent two independent colonizations resulting in two or three species, and each of these lineages group with continental *Arytaina* spp. on Genisteae hosts. The 18 *Arytinnis* species represent the largest monophyletic group resulting from a single colonization event and group with continental *Livilla* spp. on Genisteae hosts. The only island member of *Livilla*, *Livilla monospermae* Hodkinson, also groups strongly (100%) with continental *Livilla* spp. on Genisteae hosts.

*Diaphorina* is a large genus and only a relatively small sampling is included in our analysis. Nevertheless, *D. gonzalezi*, which is found on the endemic host plant, *Gymnosporia cassinoides* (Celastraceae), groups strongly (100%) with *Diaphorina gymnosporiae* Mathur, a species from South Asia also found on *Gymnosporia* spp. This is the only large-scale geographic disjunction evident for the Macaronesian psyllid taxa, all other lineages have continental relatives from near or adjacent continental regions.

The *Strophingia* species group strongly (92%) with a continental species, *S. ericae*, and all taxa feed on Ericaceae (see previous section on inter-archipelago colonizations).

The three species of *Euphyllura* are considered native (two endemic), with each representing a separate colonization event. All three species have host plants in the family Oleaceae. *Euphyllura confusa* (known only from Tenerife) groups strongly (100%) with a Mediterranean species occurring natively in Macaronesia (on Gran Canaria), *Euphyllura olivina*, and both are found on cultivated olive trees, *Olea europaea*. The short branch length and sequence divergence (cox1: 3.8%) between these species (Fig 1), as well as minimal morphological differentiation [43] suggest that *Euphyllura confusa* Bastin, Burckhardt & Ouvrard represents a relatively recent diversification on the archipelago. In contrast, *Euphyllura canariensis* on host *Picconia excelsa* is not strongly supported grouping with a particular taxon in our sampling.

The four *Agonoscena* species are all considered native (two endemic and two non-endemic), and likely each represent independent colonization events. Despite the strongly supported (92%) grouping and relatively recent divergence of *A. sinuata* (an endemic species on host plant *Ruta pinnata* (Rutaceae)) with *A. targionii* (a widespread Western Palaearctic species on *Pistacia lentiscus* (Anacardiaceae)), it seems unlikely this divergence represents an insular Macaronesian speciation event with back colonization to the continent by *A. targionii*. The

other endemic species, *A. atlantica* Bastin, Burckhardt & Ouvrard, is also strongly supported (92%) grouping, but with greater sequence divergence, with a continental species, *A. pistaciae*; and these in turn group (but with weak support) with *A. cisti*, which is another widespread Western Palaearctic species occurring natively in Macaronesia. Interestingly, the host plant for all *Agonoscena* species in our analysis except *A. sinuata* Bastin, Burckhardt & Ouvrard is *Pistacia* (Anacardiaceae), and only one other *Agonoscena* species is known from *Ruta* or Rutaceae (*A. succincta* (Heeger) occurring in the Mediterranean). Wider sampling of *Agonoscena* would clarify our interpretation, but at least one host switch from Anacardiaceae to Rutaceae has occurred with this switch possibly concurrent with colonization of the Canary Islands for *A. sinuata*; both host families are in the Sapindales.

The monotypic endemic genus *Megadicrania* is found on an endemic tree, *Olea cerasiformis* as well as on cultivated olive trees, *Olea europaea*, and is well supported (96%) within subfamily Euphyllurinae which includes other species on *Olea*, but no support for the generic placement within the subfamily was recovered.

*Lisronia echidna* on *Cistus monspeliensis* (Cistaceae) is strongly supported (92%) as grouping with *Rhinocola aceris* (Linnaeus) on *Acer* (Sapindaceae); although not included in our sampling, there are two continental *Lisronia* species in the Mediterranean region known from *Cistus* host plants, and in particular *L. varicicosta* (Hodkinson & Hollis) is considered to be the most likely sister taxon of the endemic Canarian *L. echidna* [72] and both species occur on the same widespread host plant (*C. monspeliensis*) that is found in Macaronesia, western Mediterranean, and North Africa. Therefore, although strong support grouping *Lisronia* and *Rhinocola* implies a phylogenetically wide host switch occurred between these host plant families at some point, the colonization of the Canary Islands by *Lisronia* most likely follows the common pattern of colonization of a familiar host plant.

Thus, there are few instances where the host plant association of island taxa are not readily predicted by hosts associations in continental relatives. The endemic genus *Percyella* and the genus *Drepanoza* are the only examples (see following section).

## Additional analyses to determine the origins of *Drepanoza* and *Percyella*

All three analyses of the cox1 data alone (NJ, BI, ML; Fig 2 and S1–S3 Figs) strongly support the monophyly of both *Drepanoza* and *Percyella* and provide moderate to strong support for grouping *Drepanoza*, *Percyella*, *Dyspersa* Klimaszewski, *Spanioza*, and *Hemitrioza* Crawford together as a subgroup within Group A but without consistent support for a specific generic grouping within this subgroup. *Percyella* groups moderately strongly (90%) as sister to a group with configuration (*Percyella* (*Spanioza* (*Dyspersa*, *Hemitrioza*))) in the Psylloidea backbone analysis, and in all our analyses *Percyella* and *Drepanoza* appear topologically close. One species of *Percyella* (*P. benahorita*) was included in the original mitogenome data [52] but no members of *Drepanoza* are in the backbone phylogeny, and this may contribute to the uncertainty in placing *Drepanoza* in the backbone analysis, as our placement relies entirely on the short cox1 and cytb sequences.

*Drepanoza* in the Canary Islands has two species on *Convolvulus* (Convolvulaceae) and one species on *Withania* (Solanaceae); both host families are in the Solanales. Two additional *Drepanoza* taxa on Madeira are hosted by *Pittosporum* (Pittosporaceae) in the Apiales. These different host groups for *Drepanoza* may each reflect an independent colonization of Macaronesia, but taxa on the same host family do not always group together. Within *Drepanoza*, the cox1 analyses group the two Canary Islands taxa on *Convolvulus* hosts, *D. canariensis* Bastin, Burckhardt & Ouvrard and *D. fruticulosi* Bastin, Burckhardt & Ouvrard, together with strong support (≥99%), but as with the Psylloidea analysis this is the only strongly supported

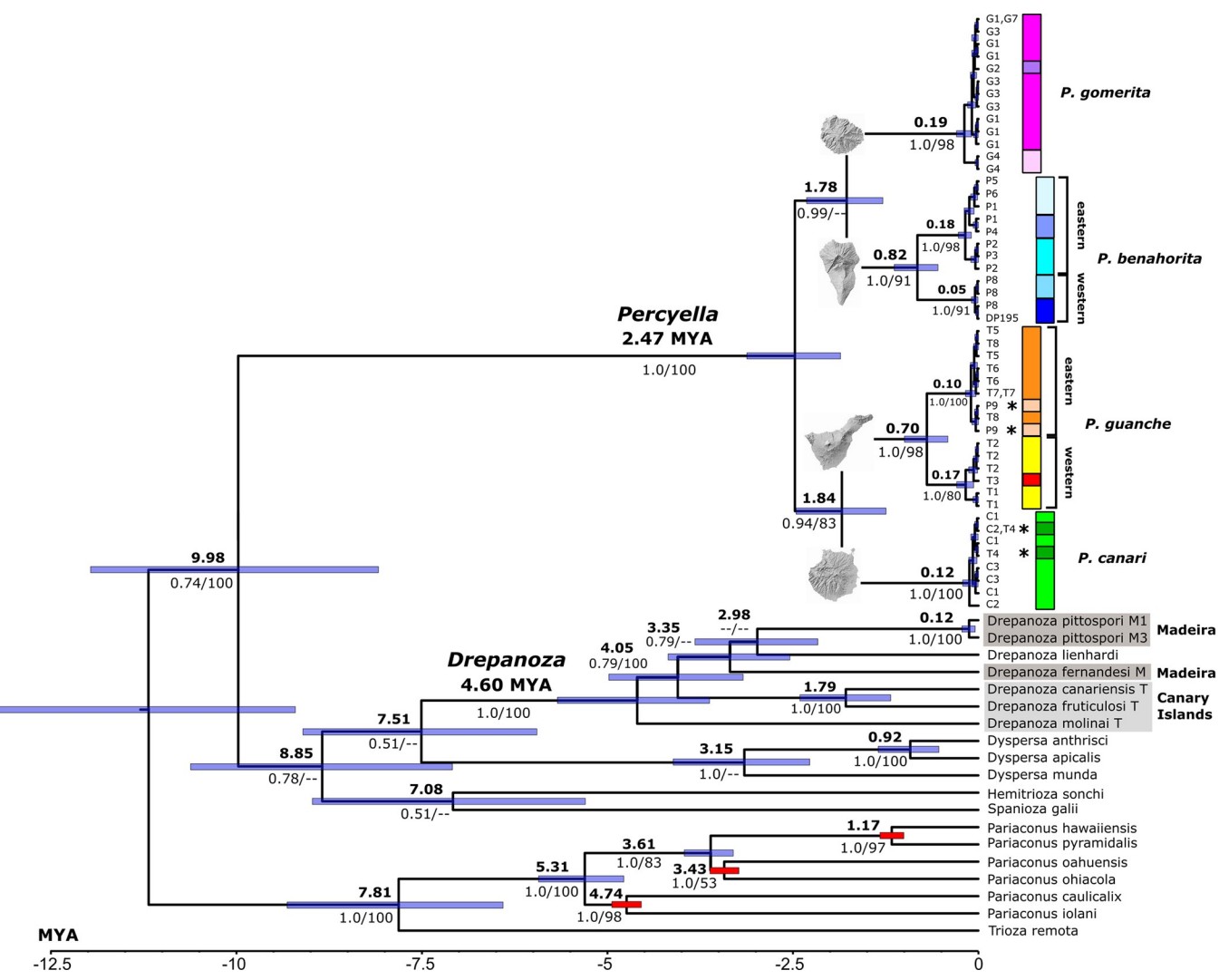

**Fig 2. Bayesian inference dating analysis using cox1 data for *Percyella* and *Drepanoza* with age priors on three nodes (red node bars) in the Hawaiian *Pariaconus* radiation in order to date the Macaronesian *Percyella* and *Drepanoza* genera (see text).** Node bars indicate 95% HPD with mean age given in bold above nodes. Support values: BI/ML are given below nodes (individual BI and ML trees are provide in S2 and S3 Figs). Vertical bars for *Percyella* species use the same colour scheme as mapped in Fig 3; asterisks indicate individuals interpreted as transported between islands. Three instances of identical sequence in *Percyella* were not included in the analysis but individual sample codes indicate their placement. *Drepanoza* consists of species endemic to Madeira (dark grey boxes), Canary Islands (light grey box), and one continental species (*D. lienhardi*).

grouping within the genus. Only the NJ analysis groups together the two Madeiran taxa on *Pittosporum*, *D. fernandesi* (Aguiar) and *D. pittospori* (Aguiar), with moderately strong support (82%), and only this analysis recovers the two Solanaceae-feeding species together, *D. molinai* Bastin, Burckhardt & Ouvrard (on *Withania*) and *D. lienhardi* (Burckhardt) (on *Lycium*), but with very weak support. Individual phylogenetic analyses (NJ, BI, ML) with support values are shown in S1–S3 Figs.

*Percyella* has diversified on an endemic Canary Island *Convolvulus* (*C. floridus*) and the sister group relationship within Group A in the backbone analysis does not reflect any particular host associations other than that the hosts in this subgroup are primarily euasterids, particularly in Asteraceae and Apiaceae. Within *Percyella*, all three cox1 analyses (NJ, BI, ML) group together with strong support the two more easterly island taxa: *Percyella canari* and *P. guanche*,

from Gran Canaria and Tenerife respectively; but only the BI analysis also groups the two more westerly island taxa together (with strong support): *P. gomerita* and *P. benahorita*, from La Gomera and La Palma respectively (Fig 2 and S1–S3 Figs). All three analyses show the striking geographic structure in *P. guanche* and *P. benahorita* with eastern and western clades within each island.

## Dating analyses and characterization of the *Percyella* island radiation on *Convolvulus floridus* (Convolvulaceae)

The two BI calibration analyses that used a single island calibration (either Kauai or La Palma) gave non credible ages older than the island age for one or more of the other noncalibrated island lineages. Using all three island calibrations within the Hawaiian Islands gave the most credible dated tree conforming to diversification events younger than the maximum age of the islands on which the diversification events occurred. The estimated ages of diversification for *Percyella* and *Drepanoza* clearly show that *Percyella* is a considerably younger group (Fig 2). Further interpretation of the dates within *Drepanoza* is hindered by the lack of phylogenetic resolution, and only the date for the diversification of *D. canariensis* and *D. fruticulosi* on *Convolvulus* in the Canary Islands is considered a notable result. In contrast, the dated analysis for *Percyella* is more revealing.

The age calibrated Bayesian analysis shows the two *Percyella* taxa on Tenerife and La Palma are older (0.7–0.82 Mya) than the taxa on La Gomera and Gran Canaria (0.12–0.19 Mya), and diversification of populations in the eastern and western clades on Tenerife and La Palma likely coincided with colonization of, and diversification on La Gomera and Gran Canaria (Fig 2). Our dated interpretation of the *Percyella* diversification is consistent with divergence between the easterly island taxa (*P. canari*, *P. guanche*) and the western island taxa (*P. gomerita*, *P. benahorita*) at 2.47 Mya occurring during the Plio-Pleistocene transition. The initial diversification of the eastern group (1.8 Mya) and marginally younger western group (1.78 Mya) is estimated to have occurred during the early Pleistocene. Similar age estimates, but a little younger, can be deduced for the initial diversification within the more westerly islands. Maximum cox1 sequence divergence within each species is given in Table 3 and shows the notably higher divergence in *P. benahorita* (4.3%) and *P. guanche* (3.1%), than in *P. gomerita* (0.9%) and *P. canari* (0.6%).

Fig 3 shows the sampling sites and haplotype assignment for the four species on four islands. On two islands, Tenerife and La Palma, occurrence of some individuals (2–4) from a different island/species (indicated with asterisks in Figs 2 and 3) suggests individuals may be transported between islands, and this is likely human mediated along with transport of the host plant for horticultural purposes. In these two instances of "non-local" occurrences, sampling locations were from planted sites (sites T4 and site P9, Figs 2 and 3), and the individuals have identical haplotypes to a native population from the originating island. The host, *Convolvulus floridus*, is native on all four islands but it is also planted as an ornamental along roadsides and in urban environments. However, we cannot conclusively rule out natural dispersal between islands. Nevertheless, despite these occurrences, we consider the status of all *Percyella* species to be single island endemics with introductions on two islands.

The haplotype network and maps in Fig 3 illustrate the same geographic structure as the calibrated Bayesian analysis (Fig 2), but with the removal of sequence length variation provides a cleaner overview of haplotype structure. The median-joining network found 12 unique haplotypes and 46 segregating sites in the 280 bp cox1 fragment. Using geotags for each sample site and a centroid georeference for each island, 63% of variation was found within populations (excluding the four "non-local"/introduced individuals would have increased within

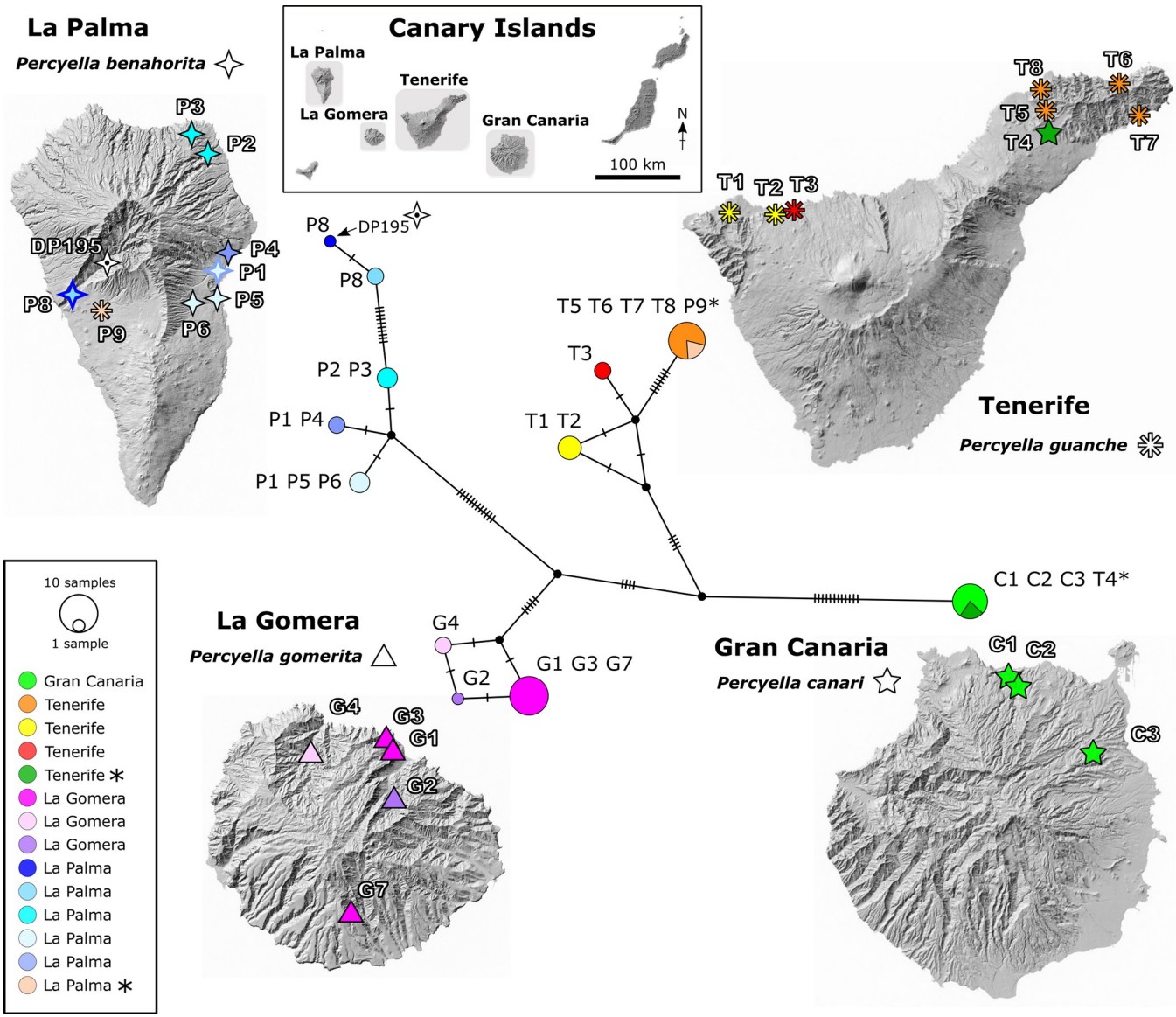

**Fig 3. Haplotype median-joining network using 280 bp of cox1 data shared by all 50 samples from four species of *Percyella* sampled for this study.** Maps show sampling locations on the four islands. The geographic location and haplotype association of sample DP195 *P. benahorita* sampled for Percy et al. [52] and represented in Fig 1 is shown. Haplotype colour coding is the same as for individuals in Fig 2. The satellite map images were obtained from source: Government of the Canary Islands (10th December 2022, https://opendata.sitcan.es/notice).

population variation). To summarize, *Percyella canari* is considered native to Gran Canaria (introduced to Tenerife) and is shown as the most homogeneous of the four species with only one haplotype despite sampling from three locations. Similarly, *P. guanche* is considered native to Tenerife (introduced to La Palma). The number of distinct haplotypes (three) is the same for *P. guanche* on Tenerife and *P. gomerita* from La Gomera, but *P. guanche* has one haplotype that is considerably more divergent from the other two and represents the eastern clade on Tenerife (also shown in Fig 2) with four sampling locations in the northeastern Anaga Peninsula (T5-T8), the other haplotypes are located in the northwest of Tenerife (T1-T3). On La Palma, *P. benahorita*, with the largest number of haplotypes (five), shows distinct divergence between three haplotypes found in the eastern part of the island (sampling sites P1-P6), and

the other two haplotypes found in the western part (sites P8, DP195), which reflects the eastern and western clades shown in Fig 2. On La Gomera, divergence of haplotypes at site G4 is more apparent in the BI analysis (Fig 2) and other analyses using the full sequence lengths (S1–S3 Figs) than in the haplotype analysis.

## Discussion

### Phylogenetic backbone analysis for placing taxa within the Psylloidea tree

Although a maximum likelihood constraint tree method is not an optimal phylogenetic approach, it provides an effective best estimate solution to place taxa with limited sequence data when a reasonably well resolved backbone is available [52,73,74]. Short fragments of fast evolving mitochondrial regions rapidly become saturated and unreliable for resolving deeper phylogenetic events and can be insufficient alone for a reliable systematic hypothesis [75]. In addition, we acknowledge that placement of taxa for which no close relatives are present in the original backbone phylogeny (e.g., for *Megadicrania*) can be problematic, and in these instances additional mitogenome data would be optimal. Despite these caveats, and partly due to sampling in the original mitogenome data containing most of the genera or close genera represented in the Macaronesian fauna, our systematic placement of most of the island taxa within the Psylloidea phylogeny are well resolved and supported, allowing interpretation of systematic placement, colonization events, and host plant associations for the majority of species.

### Phylogenetic diversity in the central Macaronesian islands and patterns of island colonization and host association

Overall, the taxonomic breadth represented in the native psyllid fauna of the central Macaronesian islands is high [43,50]. The large number of estimated colonization events (26) giving rise to the native fauna is perhaps not surprising given the islands' relative proximity to continental source areas resulting in a greater likelihood of colonization by multiple diverse lineages [39]. Consistent with general patterns of colonization into the region [1,19], most of the psyllid colonization events (69%) are represented by a single Macaronesian native species. At least five of the native genera are each represented by two to four independent colonizations events (*Agonoscena*, *Arytaina*, *Cacopsylla*, *Drepanoza*, and *Euphyllura*). Multiple colonizations within the same genus are found in other arthropod groups, examples include *Calathus* (Coleoptera) [76], *Dysdera* (Arachnida) [77], and *Sphingonotus* (Orthoptera) [78]. In these cases, independent colonizations from congeneric species are mostly confined to different archipelagos or to different islands within an archipelago if they share ecologically similar niches. Niche preemption (i.e., incumbent advantage), also known as the priority effect [79,80], is one mechanism that may promote this distribution, whereby multiple colonizations of ecologically equivalents only establish in allopatry [81,82]. Similar processes have been proposed for Macaronesian plant species [16]. Among Macaronesian psyllids, only *Strophingia* is potentially consistent with this process, but interpretation is hindered by the need for further sampling of continental *Strophingia* taxa to establish one or multiple colonizations by this genus. *Drepanoza* has independent colonizations on different archipelagos, but these are not ecologically equivalent as taxa occur on different host plant families. The remaining four genera represented by multiple colonizations can be found on the same islands within the Canary Islands; and two of these (*Agonoscena* and *Cacopsylla*) are not ecologically equivalent as the hosts are in different plant families; other genera with multiple colonizations (e.g., *Arytaina* and *Euphyllura*) can be found on related plants, but not the same host species. Interestingly, dated phylogenies for the

host plants of *Cacopsylla* and *Euphyllura* indicate the origin of the host lineages in Macaronesia are asynchronous [83,84]. Therefore, during different historical periods, priority effects may have been stronger and colonization history more relevant if both immigrants and incumbents favored similar niches (e.g., the same or closely related host plants) [82]. In summary, priority effects are not evident in observed distributions but cannot be discounted. In contrast, independent colonizations by congeneric psyllids appear mostly facilitated by ecological non-equivalence (i.e., differences in progenitor host preferences).

As with other host specific insects in Macaronesia, island psyllid lineages have been shown to exhibit efficient sequential codiversification with rapid colonization of available and familiar plant lineages [25,85]. In all but a few cases, host plant associations of island taxa are readily predicted by hosts associations in continental relatives/progenitors. More than 60% of colonizations involved use of the same host plant species, or same host plant genus as continental relatives. Notably, the island-continent species pair in *Lauritrioza* and *Diaphorina* are hosted by sister plant species [84], suggesting a common route of dispersal for insect and host plant as well as possibly contemporaneous plant-insect diversification. The Macaronesian lineages on genistoid legumes (*Arytinnis*, *Arytaina*, *Arytainilla* and *Livilla*) are all examples of host switches to related legumes [38] while the island–continent species pairs in *Euphyllura* and *Agonoscena* (and potentially *Lisronia*) are likely examples of allopatric speciation without a host switch. Only two lineages involved colonization with a host switch to a different but related plant family (*Agonoscena sinuata* and *Percyella*).

## Island distribution and extinction

Within the Canary Islands, the majority of psyllids are on the five central and western islands (Gran Canaria, Tenerife, La Gomera, La Palma and El Hierro), and in particular, the two central islands (Gran Canaria and Tenerife). Only a few psyllids have been recorded from the two drier, eastern islands (Lanzarote and Fuerteventura) and some native species records are considered unverified [43]. The five central and western Canary Islands, referred to as 'high' islands due to higher elevations that support forested habitats, are well known for species diversity, species radiations, and endemism in many plant and animal groups, including psyllids [38,86–90]. On the eastern islands, the low heterogeneity of habitats associated with the advanced eroded stage of these islands has been used to explain the low number of endemic species [91,92]. However, recent studies have suggested the islands' historical ontogeny and climatic fluctuations during the Pleistocene better explain the low number of endemic species [18,19,93]. Therefore, some psyllid species and their hosts now found only in the western islands may once have occurred in these islands before the Pleistocene extinction [93]. For instance, colonization times for *Percyella* (2.5 Mya) as well as for *Arytinnis* (2.5 Mya) and the host genus *Teline* (2.9 Mya) [25] predate the mid-Pleistocene transition (~0.8 Mya). Origins of the native psyllid lineages are almost entirely from proximal Mediterranean regions, southern Europe and north Africa which conforms with general patterns in the Macaronesian flora and fauna [16,19]. Only *Diaphorina gonzalezi* shows a remarkable disjunction with the closest relative found in South Asia. Interestingly, the Macaronesian host plant of *D. gonzalezi*, *Gymnosporia cassinoides*, is also disjunct from its closest relatives found in East Africa, Arabian Peninsula and South Asia [84], and therefore both plant and psyllid may be Tertiary relicts following on from climatic changes in the late Pliocene [94–96].

The Canary Islands has a greater taxonomic diversity and many more species than Madeira, which is not surprising given the larger number of islands and diversity of habitats. A similar pattern is found in many Macaronesian groups [86,97]. Notably, only two or three colonization events (*Arytinnis*, *Percyella*, and possibly *Strophingia*) resulted in in situ diversification of

more than two or three species, and the majority of colonization events, 18, resulted in no additional cladogenesis. No or limited species radiation may seem surprising given the varied diversity of islands and habitat types and the old age (21 Mya) of the Canary Islands, particularly when compared with the Hawaiian Islands–a much younger archipelago (5 Mya) with a similar number of islands and habitat diversity. The Hawaiian archipelago has 74 native psyllid species in 11 genera resulting from as few as eight colonization events [98]. By comparison, the central Macaronesian islands have 58 native species in 17 genera resulting from 26 colonization events. However, this disparity between Macaronesian and Hawaiian archipelagos also conforms to patterns more generally: the Hawaiian Islands has an estimated ~940 endemic species from an estimated 169 colonization events, and the Canary Islands has ~600 endemic species originating from ~230 colonization events [93,99]. Despite similar temporal diversification periods for the origin of most of the extant biotas in both regions ($\leq$ 5 Mya) [100], the combination of greater geographic isolation and climatic buffering of the Hawaiian archipelago has likely resulted in the much greater in situ diversification coupled with reduced extinction [7,98,100,101].

Dispersal limitation can be a driver of species richness [102,103] and consequently, limited cladogenesis in Canary Island psyllid lineages may result from relatively more numerous colonization events. As well as continental proximity, favorable trade winds likely increase rates of transportation of small insects like psyllids [7,27,104]. Multiple colonizations by psyllid lineages already preadapted to island plant lineages could rapidly occupy vacant ecological niches (e.g., familiar host plants), and in this way, colonizer packing fills ecological niches faster than is possible via in situ evolution. Conversely, many endemic Hawaiian species are considered to have emerged when evolution outpaced immigration as a source of novel diversity [105]. To summarize, three factors advantage colonizing psyllids over de novo species in Macaronesia: a) proximity of the Macaronesian islands to immigrant sources, b) similarity of the floras in Macaronesia and source areas, and linked to a) and b), c) preadaptation of immigrants to the same or closely related host plants [25,38,39]. Limited diversity in Macaronesian host plant lineages also determines the extent of diversity within island psyllid lineages, as psyllid radiations almost exclusively involve switching between closely related host plants [28,34,38,106]. Among psyllid host plant genera in Macaronesia, only the genus *Teline*, hosting the psyllid radiation of *Arytinnis*, has undergone substantial in situ radiation [54]. Other host plant genera have undergone little or no further diversification in the region, for example, *Chamaecytisus*, *Erica*, *Olea*, *Picconia*, *Rhamnus*, *Salix*, *Spartocytisus*, and *Withania* are each represented by only one or two endemic species (see S3 Table). However, it is worth noting that, although less commonly encountered and apparently more prevalent in particular psyllid taxa, there are instances where phylogenetically wide host switching (between plant families) is the dominant process during diversification (e.g., *Powellia* in New Zealand; see Martoni et al. [107]), and numerous instances of wide host switching have been suggested as a key process during the evolution of Psylloidea that enabled psyllids to colonize many different plant families [28]. More studies revealing the host plants of sister taxa will be needed to understand the occurrence rate of these apparently relatively rare wide host switching events in psyllids.

### *Convolvulus*-feeding and *Percyella* diversification

Given the above argument, and as two *Convolvulus*-feeding psyllid genera are already present in the Canary Islands, there is no obvious explanation why several endemic island *Convolvulus* species, which are apparently vacant niches (see S2 Table), have not been colonized by psyllids. One explanation is that Convolvulaceae is an uncommon host group for psyllids (<10 Convolvulaceae feeders worldwide) due to specific inhibitors preventing ready access to this plant

family, even to the extent of inhibiting host switches from one *Convolvulus* species or clade to another. Apart from Canary Island psyllids on *Convolvulus*, the only other confirmed host records from this plant family are in *Bactericera* (Triozidae) and *Diaphorina* (Psyllidae) [108]. Yet, there have been two independent colonizations of *Convolvulus* in the Canary Islands (by *Drepanoza* and *Percyella*). Both from genera in family Triozidae which is dominant on euasterid host groups [28], but in neither case are there known close relatives in geographical source areas that could explain Convolvulaceae as a host selection in island taxa. The closest taxa in the Psylloidea phylogeny are all on hosts in other euasterids: Apiales, Asterales, and Solanales.

The plant genus *Convolvulus* in the Canary Islands is composed of nine endemic species in two distinct clades that are from distantly related lineages with distinct morphologies, and these represent two independent colonizations from different regions of the Mediterranean [16]. The first clade includes host plants of two *Convolvulus*-feeding psyllids in *Drepanoza* witheach psyllid species occurring on a different host species (on *Convolvulus fruticulosus* and *C. canariensis*). The second *Convolvulus* clade has three species and one of these, *C. floridus*, is host to the modest radiation of four psyllid species in *Percyella*. The relatively young age of *Percyella* compared to *Drepanoza* may also partly explain why diversification on other island *Convolvulus* or indeed other euasterid hosts has not occurred in *Percyella* but has, albeit with only a single host switch on *Convolvulus*, in the older genus *Drepanoza*.

The diversification of *Percyella* in the Canary Islands is a textbook example of allopatric speciation, with a single species on each of four islands, but with no apparent ecological niche specialization because all species are on the same host plant. Genetic divergence within *Percyella* species at first glance seems contrary to expectations. *Percyella canari* is the most homogeneous of the four species, followed by increasing haplotype diversity in *P. gomerita*, *P. guanche* and maximum haplotype diversity is found in *P. benahorita*. This pattern is counter to expectations based on island age, as *P. canari* occurs on the oldest of the four islands and *P. benahorita* on the youngest. However, the dated *Percyella* radiation implies the genus is relatively young and therefore the structure and extent of diversification within each species is less a product of maximum island age, but more likely influenced by recent periods of volcanism and individual island topography. This scenario would explain the greater genetic diversity evident in the more geographically structured and isolated populations on the more geologically dynamic islands of Tenerife and La Palma. Other studies of phytophagous insects on geologically volatile islands found similarly important roles for geography. For instance, early diversification of Hawaiian planthoppers was explained by complex island topography rather than host niche specialization [109], and dynamic volcanic environments were found to be important in structuring Hawaiian spider populations [110].

## Conclusions

We present the most comprehensive phylogenetic survey of the central Macaronesian psyllid fauna to date. We provide new molecular data for 42 of the native and adventive species as well as some of the continental outgroups. We present a phylogenetic framework for understanding the origins and evolution of Macaronesian taxa, including characterization of the first psyllid radiation known on Convolvulaceae. Additionally, the molecular data provides a DNA barcode library for both native and adventive species on these islands that should prove a useful resource for evolutionary and applied research.

## Supporting information

**S1 Fig. Neighbor-joining analysis using cox1 data for *Percyella*, *Drepanoza*, and select outgroup taxa from Group A [52].** Full length sequences were used for the *Percyella* samples

(sequence length shown with sample code).
(PDF)

**S2 Fig. Bayesian analysis using cox1 data for *Percyella*, *Drepanoza*, and select outgroup taxa from Group A [52].** Full length sequences were used for the *Percyella* samples (sequence length shown with sample code).
(PDF)

**S3 Fig. Maximum-likelihood analysis using cox1 data for *Percyella*, *Drepanoza*, and select outgroup taxa from Group A [52].** Full length sequences were used for the *Percyella* samples (sequence length shown with sample code).
(PDF)

**S1 Table. Summary of all non-Macaronesian psyllid taxa for which molecular data was generated.** Molecular data: cox1: cytochrome oxidase 1, cytb: cytochrome B.
(PDF)

**S2 Table. Summary of Macaronesian endemic *Convolvulus* species [13] surveyed during this study with distribution of *Convolvulus*-feeding psyllids.** Abbreviations: H: El Hierro, P: La Palma, G: La Gomera, T: Tenerife, C: Gran Canaria, F: Fuerteventura, L: Lanzarote.
(PDF)

**S3 Table. Host plant genera of the Central Macaronesian native psyllids with the number of Canarian and Macaronesian endemic and non-endemic species [13].**
(PDF)

**S4 Table. Summary of the endemic central Macaronesian psyllid species and lineages (if in situ diversification occurred), indicating number of species per lineage, host plant, and the distribution and host plant of the closest sister group (where relationship is resolved with 80% or greater bootstrap support).**
(PDF)

## Acknowledgments

We thank Jon Martin, Antonio Aguiar, Arturo Baz, Ernst Heiss, Charles Lienhard, Bernhard Merz, Igor Malenovský, Daniel Burckhardt and Pedro Oromí for supplying specimens or making their collections of Central Macaronesian psyllids available for study. We are grateful to Heidi Viljanen (FMNH) for providing information on and access to the collection of Canary Islands psyllid species of the Finnish Museum of Natural History. We also thank Antonio González Hernández, Alfonso Peña Darias, Jonathan Molina Hernández and Ángel Francisco García Hernández for their generous assistance in collecting psyllid material in the Canary Islands and Francisco Manuel Fernandes, Isamberto Silva and Antonio Aguiar in Madeira. We are grateful to Jean-Claude Streito for providing molecular data of *Agonoscena* species. We thank Brent Emerson and Heriberto López for providing lab training, and Quentin Cronk for providing laboratory facilities at the University of British Columbia. We are grateful to Estrella Hernández Suárez for her support during the development of the project. We would also like to thank the Servicio de Sanidad Vegetal of the Dirección General de Agricultura del Gobierno de Canarias for allowing us the use of its equipment and facilities. For permits to collect, we thank the Viceconsejería de Medio Ambiente (Gobierno de Canarias) and the Parque Nacional de Garajonay (La Gomera) (registro de entrada: 6.738, TELP/249). We thankFrancesco Martoni and an anonymous reviewer for their useful comments which improved an earlier version of the manuscript.

## Author Contributions

**Conceptualization:** Saskia Bastin, Felipe Siverio de la Rosa, Diana M. Percy.

**Data curation:** Saskia Bastin, Diana M. Percy.

**Formal analysis:** Saskia Bastin, Diana M. Percy.

**Funding acquisition:** Felipe Siverio de la Rosa.

**Investigation:** Saskia Bastin, J. Alfredo Reyes-Betancort, Felipe Siverio de la Rosa, Diana M. Percy.

**Methodology:** Diana M. Percy.

**Project administration:** Felipe Siverio de la Rosa, Diana M. Percy.

**Resources:** J. Alfredo Reyes-Betancort, Felipe Siverio de la Rosa, Diana M. Percy.

**Software:** Diana M. Percy.

**Supervision:** J. Alfredo Reyes-Betancort, Felipe Siverio de la Rosa, Diana M. Percy.

**Validation:** J. Alfredo Reyes-Betancort.

**Visualization:** Diana M. Percy.

**Writing – original draft:** Saskia Bastin, Diana M. Percy.

**Writing – review & editing:** Saskia Bastin, J. Alfredo Reyes-Betancort, Felipe Siverio de la Rosa, Diana M. Percy.

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
