## [Decision Letter · Decision Letter 0]

30 Oct 2023

PONE-D-23-30814Origins of the central Macaronesian psyllid lineages (Hemiptera; Psylloidea) with characterization of a new island radiation on endemic Convolvulus floridus (Convolvulaceae) in the Canary IslandsPLOS ONE

Dear Dr. Percy,

Thank you for submitting your manuscript to PLOS ONE. After careful consideration, we feel that it has merit but does not fully meet PLOS ONE’s publication criteria as it currently stands. Therefore, we invite you to submit a revised version of the manuscript that addresses the points raised during the review process.

This paper has been deemed useful and relevant by two reviewers. Both have recommended publication with only a few minor edits. In most instances, they are simply for aspects such as readability. I expect that this will be of little difficulty to address and look forward to the revised version.

We look forward to receiving your revised manuscript.

Kind regards,

Sean Michael Prager, Ph.D.

Academic Editor

PLOS ONE

Journal Requirements:

"Funding statement: This research was carried out with financial support from the research project CUARENTAGRI (MAC2/1.1a/231)." 

"This research was carried out with financial support from the research project CUARENTAGRI (MAC2/1.1a/231)."

"Funding statement: This research was carried out with financial support from the research project CUARENTAGRI (MAC2/1.1a/231)."

6. We note that Figure 3 in your submission contain [map/satellite] images which may be copyrighted. All PLOS content is published under the Creative Commons Attribution License (CC BY 4.0), which means that the manuscript, images, and Supporting Information files will be freely available online, and any third party is permitted to access, download, copy, distribute, and use these materials in any way, even commercially, with proper attribution. For these reasons, we cannot publish previously copyrighted maps or satellite images created using proprietary data, such as Google software (Google Maps, Street View, and Earth). For more information, see our copyright guidelines: http://journals.plos.org/plosone/s/licenses-and-copyright.

a. You may seek permission from the original copyright holder of Figure 3 to publish the content specifically under the CC BY 4.0 license.  

Reviewers' comments:

Reviewer's Responses to Questions

**Comments to the Author**

1. Is the manuscript technically sound, and do the data support the conclusions?

Reviewer #1: Yes

Reviewer #2: Yes

2. Has the statistical analysis been performed appropriately and rigorously? 

Reviewer #1: N/A

Reviewer #2: Yes

3. Have the authors made all data underlying the findings in their manuscript fully available?

Reviewer #1: Yes

Reviewer #2: No

4. Is the manuscript presented in an intelligible fashion and written in standard English?

Reviewer #1: Yes

Reviewer #2: Yes

5. Review Comments to the Author

Reviewer #1: This is a well researched and well written manuscript about the evolution of psyllids in Macaronesia. I recommend therefore its publication in Plos One.

A few suggestions are written in the manuscript.

Daniel Burckhardt

Reviewer #2: Great work, it was a pleasure to read.

The figures are particularly nice, and extremely useful.

My minor comments are attached in pdf.

The "no" in regards to data availability is simply due to the lack of accession numbers, which I suspect the authors are waiting to upload in order to link this MS to the data, which is ideal.

6. PLOS authors have the option to publish the peer review history of their article (what does this mean?). If published, this will include your full peer review and any attached files.

Reviewer #1: No

Reviewer #2: **Yes: **Francesco Martoni

---

## [Author Response · Author response to Decision Letter 0]

17 Dec 2023

Please see the uploaded "Cover Letter" where we have responded to all editorial comments, and the uploaded "Response to Reviewers" document where we have responded to all the Reviewers' comments. Thank you. Diana Percy

---

## [Editor Report · Decision Letter 1]

28 Dec 2023

Origins of the central Macaronesian psyllid lineages (Hemiptera; Psylloidea) with characterization of a new island radiation on endemic Convolvulus floridus (Convolvulaceae) in the Canary Islands

PONE-D-23-30814R1

Dear Dr. Percy,

We’re pleased to inform you that your manuscript has been judged scientifically suitable for publication and will be formally accepted for publication once it meets all outstanding technical requirements.

Kind regards,

Sean Michael Prager, Ph.D.

Academic Editor

PLOS ONE
---

## [Editor Report · Acceptance letter]

18 Jan 2024

PONE-D-23-30814R1 

PLOS ONE

Dear Dr. Percy, 

I'm pleased to inform you that your manuscript has been deemed suitable for publication in PLOS ONE. Congratulations! Your manuscript is now being handed over to our production team.

Kind regards, 

on behalf of

Dr. Sean Michael Prager 

Academic Editor

PLOS ONE